# Single-molecule sequencing detection of *N6*-methyladenine in microbial reference materials

Alexa B. R. McIntyre[1,2], Noah Alexander[1], Kirill Grigorev[1], Daniela Bezdan[1], Heike Sichtig[3], Charles Y. Chiu [4,5] & Christopher E. Mason [1,6,7,8]

The DNA base modification *N*6-methyladenine (m[6]A) is involved in many pathways related to the survival of bacteria and their interactions with hosts. Nanopore sequencing offers a new, portable method to detect base modifications. Here, we show that a neural network can improve m[6]A detection at trained sequence contexts compared to previously published methods using deviations between measured and expected current values as each adenine travels through a pore. The model, implemented as the mCaller software package, can be extended to detect known or confirm suspected methyltransferase target motifs based on predictions of methylation at untrained contexts. We use PacBio, Oxford Nanopore, methylated DNA immunoprecipitation sequencing (MeDIP-seq), and whole-genome bisulfite sequencing data to generate and orthogonally validate methylomes for eight microbial reference species. These well-characterized microbial references can serve as controls in the development and evaluation of future methods for the identification of base modifications from single-molecule sequencing data.

[1] Department of Physiology and Biophysics, Weill Cornell Medicine, New York 10065 NY, USA. [2] Tri-Institutional Training Program in Computational Biology and Medicine, New York 10065 NY, USA. [3] US Food and Drug Administration, Silver Spring 20993 MD, USA. [4] Department of Laboratory Medicine, University of California San Francisco, San Francisco 94107 CA, USA. [5] UCSF-Abbott Viral Diagnostics and Discovery Center, San Francisco 94107 CA, USA. [6] The HRH Prince Alwaleed Bin Talal Bin Abdulaziz Alsaud Institute for Computational Biomedicine, Weill Cornell Medicine, New York 10021 NY, USA. [7] The Feil Family Brain and Mind Research Institute, Weill Cornell Medicine, New York 10021 NY, USA. [8] The WorldQuant Initiative for Quantitative Prediction, Weill Cornell Medicine, New York 10021 NY, USA. Correspondence and requests for materials should be addressed to C.E.M. (email: chm2042@med.cornell.edu)

N6-methyladenine is the most common modified base in bacterial DNA and plays roles in restriction modification (RM) systems[1], new strand DNA repair[2], and regulation of gene expression[3,4]. Methyltransferase mutations and over-expression can affect virulence[5,6] and interactions between bacteria and host cells[7]. No enzymes are known to demethylate DNA m6A in vivo[8] but competitive protein binding at target sites can heritably affect methylation in newly replicated cells and thus modulate transcription[4]. Frame-shift mutations at repeat regions in methyltransferase genes can also lead to changes in the expression of genes involved in cell wall formation and repair pathways[9,10]. These phase variation systems can help bacteria evade mammalian host defenses[11,12], but it is unknown whether methyltransferase frame-shifts contribute to adaptation to other environments.

Methyltransferases couple with restriction enzymes to form RM systems that protect bacteria through the selective recognition and cleavage of unmethylated motifs in foreign DNA while the same motifs in bacterial genomes are shielded by methylation. RM systems differ in target site characteristics and whether their restriction enzymes and methyltransferases are consecutively or separately encoded, which may affect their efficiency[13,14]. Type I target sites are typically symmetrical sequences separated by six to seven arbitrary nucleotides for a total length of ten to sixteen bases, type II recognition sites often palindromic and between four and six bases, and type III sites asymmetrical and between four and six bases in length. Unlike types I and II, type III methyltransferases catalyze methyl group addition to only a single strand. Based on known motifs, near homologs among types I and III methyltransferases target the same sites[14]. However, not all methyltransferases are associated with active restriction enzymes; Dam in *Gammaproteobacteria* and CcrM in *Alphaproteobacteria* play roles in replication repair and the cell cycle, respectively[8].

Prior to the development of single molecule sequencing, methods to detect m6A in DNA included restriction digests or sequencing after immunoprecipitation. Immunoprecipitation can reveal methylated areas, but not identify individual bases, while restriction enzyme-based methods are limited to particular motifs and many are unable to differentiate between hemimethylation and symmetrical methylation of both strands[15]. Alignment of data from single molecule sequencers and comparison of kinetic data to a model or an unmodified reference can more precisely localize modifications. The single-molecule, real-time (SMRT) analysis pipeline for PacBio sequencing data uses a *t*-test to compare DNA polymerase kinetics surrounding methylated bases and models for unmethylated bases, measured as an inter-pulse duration (IPD)[6,16–20]. Multiple groups have also found that small changes to electrical signal as DNA travels through nanopores can reveal the presence of base modifications using sequencers developed by Oxford Nanopore Technologies (ONT)[21–23]. Methods of detection have so far focused on common mammalian modifications, specifically looking at the eukaryotic contexts of homo-methylated CG-repeats[21] and 5-methylcytosine (m5C) versus 5-hydroxymethylcytosine (hm5C)[22], and have also been used to investigate other modifications, including m6A, albeit at lower accuracy (~70% based on published reports[22,23]). However, there are no available reference materials that have been analyzed with multiple single-molecule sequencing methods for evaluating the accuracy, validity, or detection limits of modified bases.

To address this need, we develop a method for detecting modified bases in ONT data, and generate PacBio, ONT, and Illumina data from a set of reference strains to complete de novo assemblies and methylation profiles for these strains. Specifically, using known motif sites confirmed with orthogonal PacBio data generated for ONT-sequenced *E. coli* DNA samples[24], we identify sets of methylated and unmethylated positions in the *E. coli* MG1655 K12 genome to train and test binary classifiers for the detection of m6A in nanopore data based on deviations between observed and expected currents. To further validate the best-performing classifier, we generated ONT, PacBio, and MeDIP-seq data to identify and compare detection of m6A in individual strains from a commercially available microbial reference community (ZymoBIOMICS) that includes five gram positive bacteria (*Bacillus subtilis, Enterococcus faecalis, Lactobacillus fermentum, Listeria monocytogenes*, and *Staphylococcus aureus*), three gram negative bacteria (*Escherichia coli, Pseudomonas aeruginosa*, and *Salmonella enterica*), and two fungal species (*Cyptococcus neoformans* and *Saccharomyces cerevisiae*).

## Results

**Modeling m6A effects on nanopore current.** We first demonstrate that the effects of DNA methylation on template strand sequence Phred-like quality scores vary depending on the version of the technology and base caller used, and that there is no consistent decrease in quality at modified bases (Supplementary Figure 1). ONT provides an estimated mean and standard deviation for currents corresponding to individual 5-mers or 6-mers, which were used for base calling with hidden Markov models[25], before ONT switched to recurrent neural network algorithms. These distributions assume that five or six bases in and around the pore affect the current through the pore at a given time point. Previous modification callers built similar distributions for methylated 6-mers, although Rand et al.[22] noted that these distributions were less distinguishable for m6A than m5C.

Across the extended sequence context surrounding an individual methylated site, shifts from the model differ in magnitude and direction depending on context (Fig. 1a, b). We looked at current deviations (observed-expected values) using a sliding window approach, in which each window slides over the six 6-mers composing an 11-mer surrounding an adenine. We were initially concerned that longer sequence contexts would require prohibitively large and diverse datasets for training, since the number of 11-mers that contain a single central modified position exceeds a million ($4^{10}$). Nevertheless, we hypothesized that patterns in current shifts would be shared enough to predict m6A; overall for R9 *E. coli* data, methylation at the 4th and 5th positions of a 6-mer in particular tended to increase the current with respect to the model values (Figure 1c). We thus used current deviations as features to train four binary classifiers (Fig. 2a), including neural network, random forest, naïve Bayes, and logistic regression.

The neural network classifier produced the highest accuracy, although a random forest model performed comparably (Supplementary Figure 2). Tested on a second data set from the same *E. coli* strain produced in a second lab, the model achieved 81.3% accuracy (compared to 80.8% for the random forest model) using all quality levels of reads and comparing methylated positions to a random selection of unmethylated sites in the same genome (Fig. 2b, Supplementary Table 1). The Spearman correlation between the probability estimates from the top two predictors, neural network and random forest, was high, at 0.93 (Supplementary Figure 2D). A receiver operator characteristic curve showed the changes in accuracy at varying thresholds for classification (Fig. 2c). Accuracy improved to 84.2% for higher quality reads (mean quality > 9) and decreased to 77.8% with a maximum of two skips per prediction, or 6-mers for which the sequencer missed recording a current value. When summarizing predictions at single sites with a minimum of 15× coverage, the classifier achieved 95.4% accuracy and an area under the curve (AUC) of 0.99, with comparison to true negatives drawn from

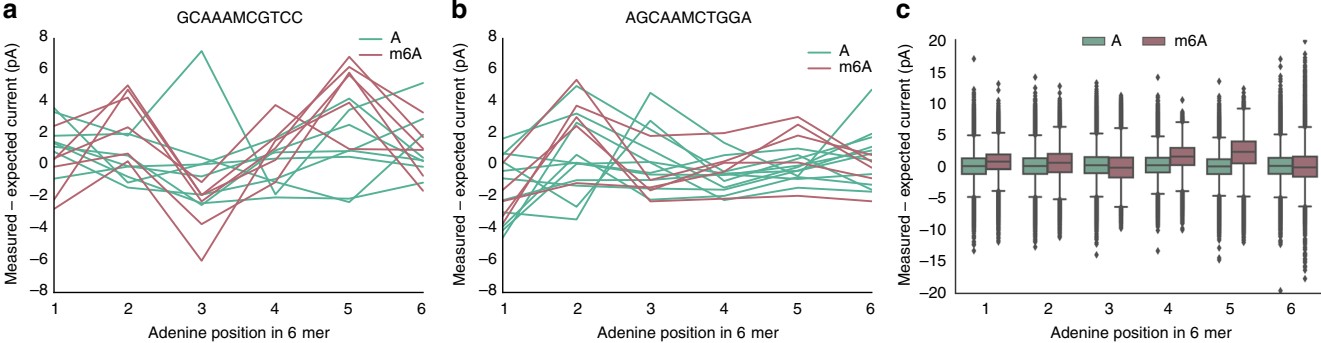

**Fig. 1** m6A methylation affects nanopore signal. Picoampere currents deviate from model values as the DNA surrounding a methylated adenine is pulled through a nanopore. **a**, **b** The deviations vary according to the position of the adenine within the pore and its surrounding sequence context. **c** Across all sequence contexts, the greatest deviations for R9 data occurred with the adenine in the fourth or fifth position among six nucleotides considered by the model in and around a pore. Boxplot center lines show medians and whiskers 1.5× interquartile range. Outliers are truncated at $+/-20$ pA to better visualize data trends

unmethylated positions, although these estimated accuracies did not account for bias towards specific sequence contexts, as discussed below. We then tested the hypothesis that methylation would affect a similar range of surrounding current levels as the canonical bases in the ONT models (six) and found that using four or eight 6-mers surrounding a base reduced classification accuracy (Supplementary Figure 2A and C).

We further tested mCaller on a second base modification found in more variable contexts, m5C. Using data from samples lacking methylation (through PCR amplification) and samples methylated using the bacterial methyltransferase M. SssI from Simpson et al.[21], we trained and tested the method for the identification of 5-methylcytosine in CG contexts. We again found features sufficiently similar across contexts for prediction, with per-read accuracy of 82.2% (Supplementary Figure 3).

**Validating reference material RM targets**. For bacterial species like *Escherichia coli* in which one enzyme (Dam) is responsible for most methylation and specifically targets GATC motifs, similarity among training sequence contexts could bias a model. We used PacBio sequencing for seven of eight bacteria (Table 1) and one of two fungi in the ZymoBIOMICS Microbial Community Standard to evaluate the accuracy of an *E. coli*-trained model for R9.5 nanopore data in a wider variety of contexts. Nineteen motifs were predicted based on the PacBio data and the overlap was taken between motif sites and MeDIP-seq peaks (Fig. 3a); previous base modification detection from nanopore data has shown variable success depending on motif[23]. To detect modifications at these sites using per-position, multi-read predictions from mCaller, we trained an **A**G model using G**A**GNNNNNTCTT sites in the Zymo *E. coli* strain and left **A**A, **A**C, and **A**T sites pooled under a separate model, which led to sufficiently clear enrichment of most target motifs to confirm their identification. For *Bacillus subtilis*, in which we annotated a type I RM system with perfect sequence homology to that of the nearly identical strain *Bacillus subtilis* T30 (99.99% genome pairwise identity)[26], we compared enrichment for 1032 unique methyltransferase target sequences from REBASE[27] using AME[28]. The test returned a corrected p-value for the known motif of this RM system (CNC**A**NNNNNNNRTGT/AC**A**YNNNNNNNTGNG, one-tailed Fisher's exact test $p = 1.56e-154$) over a hundred orders of magnitude lower than that of the next most significant motif (CG**A**YNNNNNNNRTRTC, $p = 6.18e-13$), showing the

detection of known m6A motifs is possible with nanopore data alone. Across species, calculations of percent m6A in the reference species based on motif sites detected as methylated by PacBio correlated highly (Spearman $\rho = 0.97$, $p = 6.54E-6$) with experimental values measured using an ELISA kit (although the assay appeared to consistently overestimate m6A levels), further increasing our confidence that most motifs were correctly identified (Fig. 3b).

After annotating putative methyltransferase genes in the ten strains, we looked for evidence of phase variation by scanning for simple sequence repeats according to previously described criteria[10]. We found widespread homopolymeric tracts (Table 1). Most were six to eight bases and shorter than repeats previously connected to phase variation in *Helicobacter pylori* methyltransferases (10–13 bases[11]), and involved A/T homopolymers, which were associated with weaker candidates for true phase variation in *Neisseria gonorrheae*[10]. In *Pseudomonas aeruginosa*, we detected neither methylation using ELISA nor methylated motifs using PacBio and nanopore data. We found no length polymorphisms for a five GC-dinucleotide repeat in the putative *P. aeruginosa* methyltransferase gene across 132 strains aligned using BLAST, suggesting phase variation does not explain the lack of methylation observed in our strain. Notably, the only region annotated as a potential "Adenine specific DNA methylase Mod" using RAST lacked either of the canonical catalytic motifs for amino methyltransferases, DPPY, found in 1262 of 2065 (61.1%) of unique adenine methyltransferase sequences in REBASE, and NPPY, found in an additional 680 (32.9%), providing further evidence that this reference species lacks a functional m6A methyltransferase.

We next combined mCaller with PacBio detection methods and MeDIP-seq to refine predictions of methyltransferase fidelity. Both Mod (methyltransferase) and Res (restriction) enzymes are thought to be highly specific, although there are reports of off-target effects with overexpression[29]. In cases where a Mod enzyme is linked to a nonfunctional Res enzyme, methylation is estimated to occur at more variable contexts or at a lower fraction of sites[30,31], and a non-specific *E. coli* methyltransferase has recently been described[32]. Previous analyses of methyltransferase target sites based on PacBio data report high, but not complete, methylation of target motifs for most methyltransferases, along with limited off-target effects[16,31]. In each of the strains analyzed, adenines marked as methylated by PacBio that did not correspond to motifs were called as methylated by mCaller only

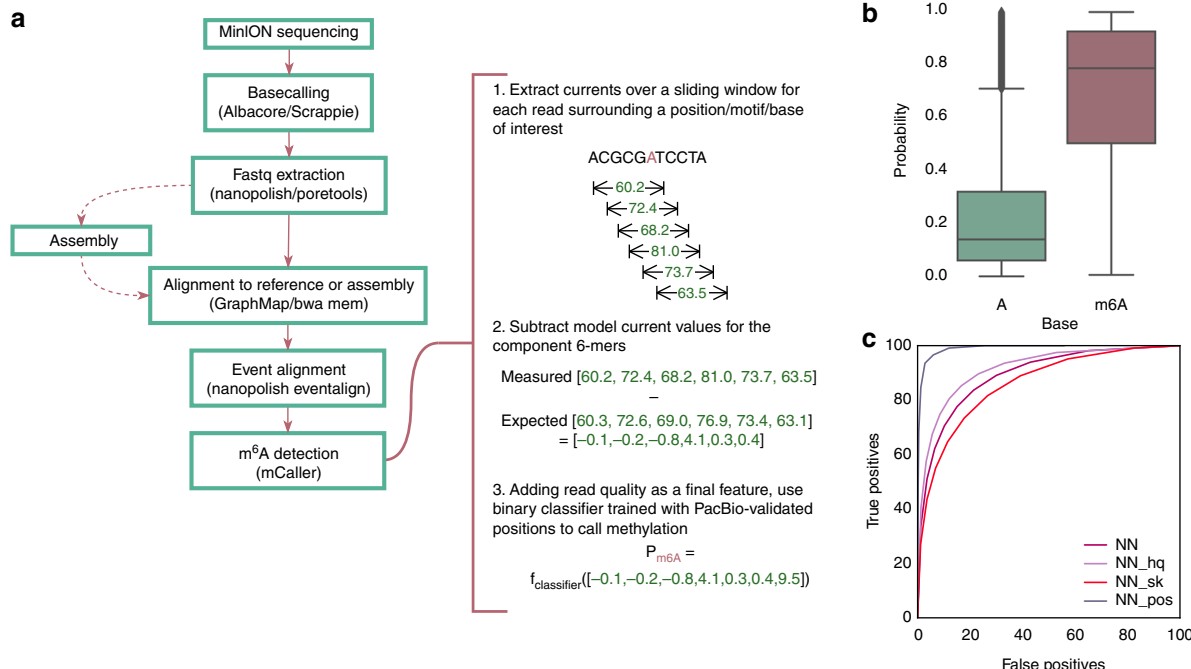

**Fig. 2** mCaller workflow and classification of *E. coli* sites in R9.4 data. **a** The pipeline for classification of adenines as methylated or unmethylated.
**b** Probabilities of methylation defined by a neural network classifier for methylated compared to unmethylated positions in *E. coli*, with a model trained on one dataset and tested on a second. **c** ROC curves for the neural network model using R9.4 data, showing true positive rate (methylated positions correctly identified) as a function of false positive rate (unmethylated positions called as methylated) with varying probability thresholds for classification. We tested modifications to the standard model ("NN") using only high quality reads (average base quality > 9, "NN_hq") and classifying observations that included a maximum of two skips ("NN_sk"). A curve was calculated for genomic positions with ≥15× coverage by varying the fraction of reads with probability of methylation scores ≥0.5 required to define a position as methylated ("NN_pos"). Boxplot center lines show medians and whiskers 1.5x interquartile range

at low rates (~10–20%) and most did not overlap with MeDIP-seq peaks (Fig. 3c). Because MeDIP-seq does not reach single base resolution, a higher percentage of peaks in strains with more m6A (*E. coli* and *S. enterica*) also overlapped non-motif sites. The low identification rates from mCaller and MeDIP-seq suggest that most non-motif sites are false positives, and indeed the low average IPD ratios and reproducibility from PacBio data support this result (Supplementary Table 2).

Before nanopore sequencing, no single-molecule sequencing validation was available for PacBio predictions of m6A motifs. For questionable motifs and motifs identified as adenine-methylated at <90% of sites by PacBio, we plotted nanopore signal changes surrounding motif sites and compared to the same motifs in *P. aeruginosa* as a negative control (Supplementary Figure 4). Smaller deviances from model values were seen in nanopore current measurements for BATGC**A**TV motifs missed by PacBio than detected in *S. enterica*, suggesting incomplete methylation at this motif and potentially a role independent of RM. By contrast, we found a consistent decrease with adenine at the 5th position for G**A**TCGVNY in *S. aureus*, indicating PacBio may underestimate modification at this motif. With PacBio, results varied between our two sequencing runs and depending on the SMRT Tools parameters used, with modifications at the GATCGVNY motif and similar motifs alternately identified as m6A, m4C, unknown A, or unknown T, but it was unclear which base was modified. To test which base was modified, we used MeDIP-seq, which did not detect high levels of m6A at these sites (30 out of 399 fell under peaks), and bisulfite sequencing, which did detect cytosine methylation (388 out of 399 sites had coverage of ≥100× and ≥50% methylation). However, standard bisulfite sequencing does not distinguish well between m4C, m5C, and hm5C, therefore we cannot confirm which cytosine modification is present[33].

Most TTA**A**NNNNNNNTAGA motif sites in *S. aureus* were not called by mCaller, but changes from model values were comparable to those for its reverse complement, TCT**A**NNNNNNNTTAA, indicating m6A is likely present at TTA**A**NNNNNNNTAGA. In *E. faecalis*, the putative motifs CTCC**A**G and CTKV**A**G showed little change in signal at any position in sites identified or missed by PacBio but MeDIP-seq suggested partial methylation of CTCC**A**G and CTKV**A**G motifs. Positions identified as m6A by PacBio were significantly more likely to fall under MeDIP-seq peaks than positions missed by PacBio (CTKV**A**G two-tailed Fisher's exact test odds ratio = 4.35, $p = 8.2e^{-117}$), suggesting these are true m6A motifs.

We next compared mCaller performance to the m6A model for Tombo, an updated version of the tool developed by Stoiber et al.[23] and the only other tool with an m6A model currently available (Fig. 4). Results varied for both tools depending on motif, with mCaller generally increasing accuracy for trained motifs (G**A**TC AUC of 0.91 for mCaller vs. 0.83 for Tombo, AAG**A**NNNNCTC AUC of 0.71 for mCaller vs. 0.86 for Tombo, and G**A**GNNNNNTCTT AUC of 0.90 for mCaller vs. 0.64 for Tombo), with true positives from the species of interest and true negatives consisting of the same motifs in the *P. aeruginosa* genome. Tombo called similarly few CTKV**A**G and CTCC**A**G sites in *E. faecalis* as m6A as mCaller (1265 vs. 1167 of CTKV**A**G sites and 78 vs. 89 CTCC**A**G sites detected as methylated by PacBio had ≥50% m6A for Tombo and mCaller, respectively), perhaps because of only modest deviations in current (Supplementary Figure 4).

For two species, *L. fermentum* and *C. neoformans*, we did not have PacBio data and our ELISA indicated low levels of m6A (Supplementary Figure 5). We ran mCaller and AME but were unable to detect any known motifs close to the level of certainty seen for *B. subtilis* ($p < 1e^{-100}$), suggesting any methylation in

| Table 1 Predicted DNA methyltransferase genes and their features in reference community bacteria | | | | | |
|---|---|---|---|---|---|
| Species (PacBio data) | Positions | Protein length | Predicted type | Short sequence repeats | Canonical catalytic motif |
| B. subtilis (Y) | 446228–448240 (+) | 671 | I | A8, A6 | NPPY |
|  | 2545852–2546838 (−) | 329 | – | A6 * 5, T6 | – |
| E. faecalis (Y) | 296156–297748 (+) | 531 | I | A7 * 4, A6 * 2 | NPPY |
|  | 1483533–1484801 (+) | 423 | – | A6 * 4, T6, A7 | DPPY |
|  | 1913501–1914508 (−) | 336 | – | A6 * 2, T6, A7, TGC4 | – |
|  | 1942230–1943087 (−) | 286 | II | T6, A6 * 2, A7 | – |
| E. coli (Y) | 2009620–2009922 (−) | 101 | Repair | A6 | – |
|  | 3996959–3997795 (+) | 279 | Repair | T6, A6 * 2 | DPPY |
|  | 4100327–4101221 (−) | 297 | II | A6, A7 | DPPY |
|  | 4759457–4761004 (−) | 516 | I | – | – |
| L. fermentum (N) | 740224–742155 (+) | 643 | III | T6 * 2, A6 * 4 | DPPY |
|  | 1665653–1666522 (−) | 289 | – | G7 | – |
| L. monocytogenes (Y) | 1259612–1260610 (+) | 333 | – | A6 | – |
|  | 2539806–2542382 (−) | 859 | I | A7 * 2, A6 * 6 | NPPY |
| P. aeruginosa (Y) | 4062465–4062764 (−) | 100 | – | GC5 | – |
| S. aureus (Y) | 939806–941362 (+) | 519 | I | A6 * 4 | NPPY |
|  | 942659–943006 (+) | 116 | – | A8, A7, A6, T7 | – |
|  | 943165–943512 (+) | 116 | – | T6, A6 | DPPY |
|  | 2353109–2351553 (−) | 518 | I | A6 * 4 | NPPY |
| S. enterica (Y) | 424176–424829 (−) | 218 | – | – | NPPY |
|  | 2945089–2945925 (+) | 279 | Repair | A8, T6 | DPPY |
|  | 3025834–3026718 (−) | 295 | II | A6 * 5 | DPPY |

these strains occurs at motifs yet to be characterized. No MeDIP-seq peaks were identified for *L. fermentum*. Two thousand eight hundred and twenty MeDIP-seq peaks were identified for *C. neoformans*, 1092 of which overlapped with at least one mCaller-predicted site (conversely, only 6527 of 213,478 mCaller-predicted sites fell under MeDIP-seq peaks). De novo motif predictions using HOMER for sites detected by both mCaller and MeDIP-seq showed enrichment of repeat regions, consistent with known noise in MeDIP-seq data[34,35]. Since false positive sites from mCaller outnumber true positives in species with trace levels of methylation, we do not yet recommend using nanopore data alone to predict new motifs.

**Methyltransferase target motifs in associated phages**. We then examined the types and distributions of RM motifs in these strains. Palindromic type II restriction modification sites are avoided in both phages[36–38] and their bacterial hosts[13]. Other types of sites are rarely depleted in hosts, but have been hypothesized to be depleted in phages[39]. We found under-representation of G**A**TC, B**A**TGC**A**TV, and C**A**G**A**G in *S. enterica* and *E. coli* genomes compared to expected values calculated by a maximum order Markov model[40] (Fig. 5a), and only slight underrepresentation of other motifs. Motif depletion in predicted prophage sequences weakly followed the same trend as in bacterial genomes, with little evidence for systematic depletion of longer, gapped, type I motifs (Fig. 5b). A second method of estimation that accounts for biases in gapped sub-motifs showed less variance and no underrepresentation of non-palindromic sites[41] (Supplementary Figure 6). Since there were a limited number of prophage sequences in each genome, we also compared to phages known to infect the same reference species[42]. We found results noisy for the palindromic targets in our dataset but greater evidence for depletion of type II targets for species-motif pairs from REBASE (Fig. 5c, d). Whether the phages associated with different species in the database could infect our specific reference strains is difficult to establish, given that host range is poorly characterized for most phages[43], and the suggestion of greater efficiency for type II RM systems would need to be confirmed in strains with both well-validated motifs and virus associations.

## Discussion
For bacteria and other kingdoms of life, single molecule sequencing with PacBio or ONT can reveal both genomic and epigenomic states of nucleic acids. Recent research showed that nanopore sequencing can detect m5C with per-read accuracy of upwards of 80% but m6A with lower accuracy (~70%). We show here that m6A, the most common bacterial base modification, can be detected using picoampere-scale changes in current as individual DNA molecules travel through a nanopore. Our program is the first designed for training using orthogonally validated positions, rather than artificially methylated or unmethylated reads. It also does not rely on the inclusion of a particular k-mer context within the training dataset to make a prediction. Stoiber et al.[23] required 20× coverage and both native DNA and whole-genome amplification of the same sample to approximately localize base modifications with predictions based on statistical differences in currents, although we note that the updated version of this tool (Tombo) shows improved resolution and no longer requires amplification. As described for Stoiber et al.'s original tool[23], accuracy varied by sequence motif; mCaller showed 60–98% per-position accuracy at most motifs, with notable failure at one *S. aureus* motif. Tombo and mCaller both showed variable performance across the motifs tested and predicted lower methylation than the combined predictions from PacBio and MeDIP-seq where current deviation were small (CTKV**A**G and CTCC**A**G motifs in *E. faecalis*). For mCaller, training biases could decrease accuracy at contexts with lower similarity to the training examples. Assuming Tombo still uses the same strategies as the 2016 version, lower accuracy than mCaller at certain motifs could indicate instances where Tombo is less successful at integrating information from multiple points in a base's passage through the pore. Overall, our results demonstrate a need for tool evaluation at a variety of sequence contexts, for which we propose the continued use of this well-validated microbial reference community.

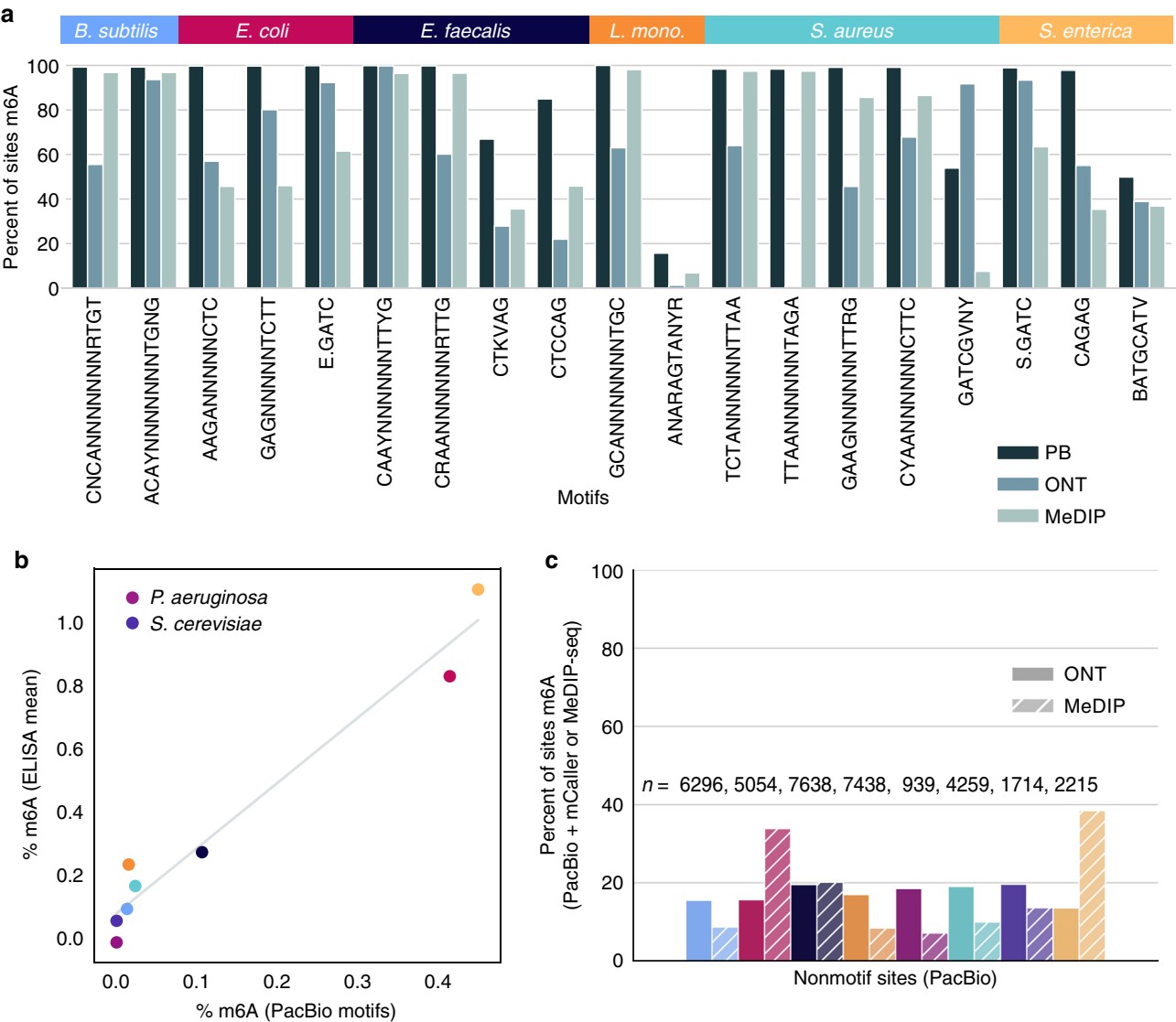

**Fig. 3** Motif detection in a microbial reference community. **a** The percent of sites at different motifs identified as methylated by PacBio and nanopore (ONT) across species in a reference community. **b** A comparison between the percent m6A in the genome detected using a commercial ELISA kit and based on the motif sites identified using PacBio and confirmed by MeDIP-seq (Spearman $\rho = 0.97$, $p = 6.54E-6$). **c** The percent of non-motif sites identified by PacBio that were also called called as methylated using mCaller. The total non-motif sites called by PacBio for each strain is noted as $n$

In most cases, mCaller could identify m6A at untrained contexts in reference strains with sufficient accuracy to find known motifs or confirm new motifs predicted through PacBio sequencing. The same non-motif sites were not often detected by the two methods, suggesting that these are false positives that reflect different sources of noise for nanopore current deviations and PacBio IPD ratios. However, both single-molecule sequencing methods can fail to distinguish between types of methylation, as we saw for the GATCGVNY motif in *S. aureus*. As previously demonstrated for PacBio sequencing of m5C following Tet conversion to 5-carboxylcytosine[44], chemical or enzymatic treatments may be able to facilitate the identification and differentiation of base modifications difficult to pick up with nanopore data. Whereas PacBio is more suitable for detecting m6A and m4C than m5C, the latter seems more easily distinguishable than m6A in nanopore data based on results thus far. Nevertheless, by combining the two data types, we identified target motifs for eight likely m6A methyltransferases in five

species and confirmed a known motif in *B. subtilis*; there was no evidence for an active adenine methyltransferase in *P. aeruginosa*. We were unable to confidently assign motifs or identify sites consistently detected using PacBio, nanopore, and MeDIP-seq data in the two *Dikarya* fungi in the reference community, *C. neoformans* and *S. cerevisiae*. Mondo et al.[45] were likewise unable to confirm PacBio-identified sites in *Dikarya* using mass spectrometry and m6A immunoprecipitation. Our ELISA results suggest that if m6A is present in either species, it is at levels far below those recently described for early-diverging fungi[45].

In bacteria, evolutionary selection against motifs associated with type II RM systems[13,36–38] and CRISPR[46] can enable phage evasion of bacterial defenses, but our results for the reference community strains and others suggest limited selection against motifs associated with type I RM systems. This suggests the possibility for faster evolution on both sides with type II RM systems, as small variations in sequence for type II but not type I or III methyltransferases lead to different target motifs[14]. Just as

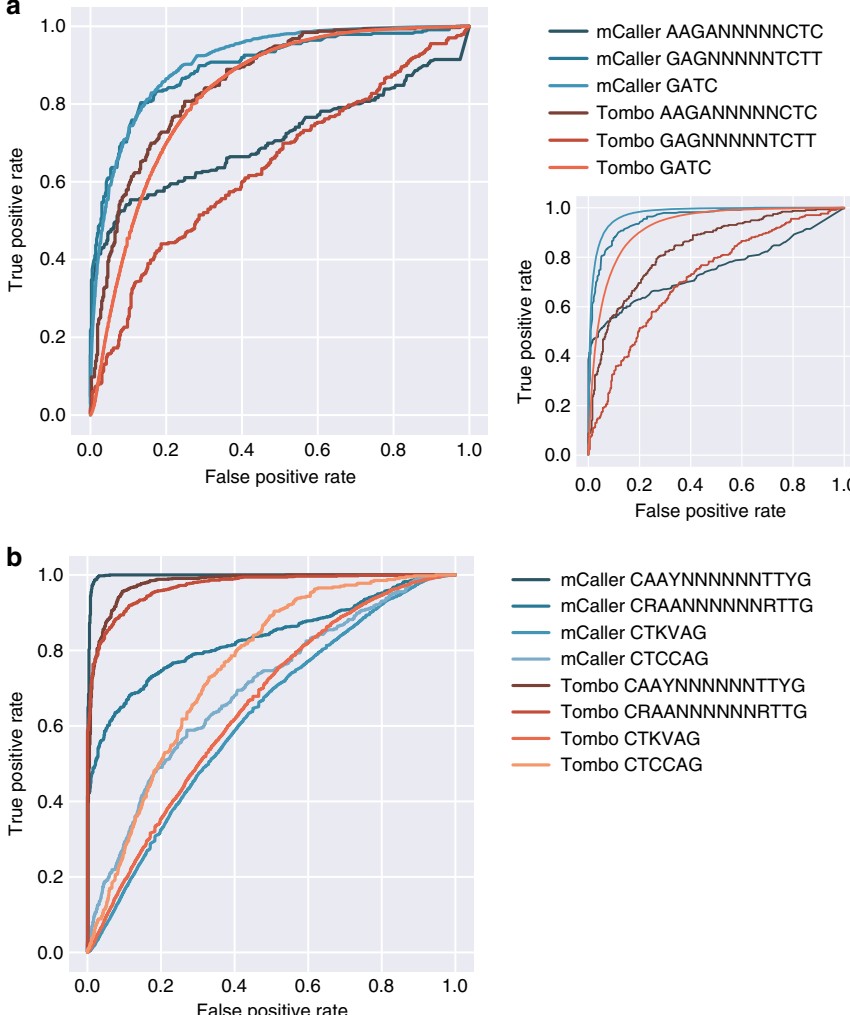

**Fig. 4** Comparison to Tombo. **a** Precision-recall curves for *E. coli* motifs called by mCaller (per-position calls, for comparison to Tombo) and Tombo for a model trained using *E. coli* data, with true negatives corresponding to the same motifs within *P. aeruginosa*. Inset: comparison to true negatives drawn from a random set of true unmethylated motif positions within *E. coli* suggests GATC motifs are more frequently identified as methylated by both mCaller and Tombo than random adenines. **b** Precision-recall curves for untrained motifs in *E. faecalis*, with true negatives again drawn from *P. aeruginosa*

motif avoidance in bacterial genomes relates to RM system lifespan[13], avoidance in phage may depend on how long a phage species has been exposed to a particular methyltransferase. Of note, a motif with incomplete methylation in *S. enterica* (BATGCATV) was still underrepresented in the host genome, as was the Dam target motif GATC in *E. coli*, suggesting that underrepresentation of ungapped palindromic motifs may not depend on current involvement in type II RM systems[38].

Although much of bacterial epigenomic research focuses on species relevant to human disease, the power of m6A to modulate gene expression suggests methyltransferase gene evolution could be a function of environmental pressures in addition to phage exposure. Recent use of nanopore sequencers during missions on the International Space Station[24], the Ebola virus outbreak in West Africa[26,27] and the Zika virus outbreak in the Americas[28,29] all demonstrate the possibility for remote genomic and epigenomic studies. Notably, our method for base modification detection can improve detection compared to existing models, depending on motif. While this method still has limitations, particularly for sequence contexts on which a model hasn't been trained, m6A methylomes of diverse reference species readily available for re-sequencing[47] can aid in training and testing of

future methods and serve as positive controls or titrated standards for metagenomic and microbial studies.

## Methods

**Model training data and parameters**. Two datasets for the same mixture of equal masses of native mouse and *E. coli* MG1655 K12 DNA, and m6A-free λ phage DNA recently sequenced by Castro-Wallace et al.[24]] were generated using version R9 flow cells and 2D kits at Weill Cornell Medicine (Mason lab) and University of California, San Francisco (Chiu lab). Reads were basecalled using Metrichor from ONT (v 2.42.2). One R7.3 run from Castro-Wallace et al.[24] was used to compare quality scores in Supplementary Figure 1. Orthogonal PacBio data for the same *E. coli* strain was also generated and a single contig was assembled using HGAP v2. Reads were realigned to the de novo assembly and methylated positions identified through differences between measured and expected inter-pulse durations using SMRTPortal v2.3.0-RS_Modification_and_Motif_analysis.1[18,19]. All methylated positions included for training were associated with known methyltransferase target motifs for the strain (GATC and AACNNNNNNGTGC/ GCACNNNNNNGTT), and had PacBio QV scores ≥20 (*p*-values ≤ 0.01) for confidence in the modification type ("identificationQv") and estimated modification fraction >90%. We excluded any sites that overlapped the λ phage genome to avoid any ambiguity in the provenance of reads (failing to do so decreased cross-validation accuracy by approximately 5%). A control set of the same number of positions was chosen at random from adenines not called as methylated at any score and not associated with *E. coli* K12 methyltransferase target motifs. Sites from the first half of the genome were used for training, and sites from the second half of

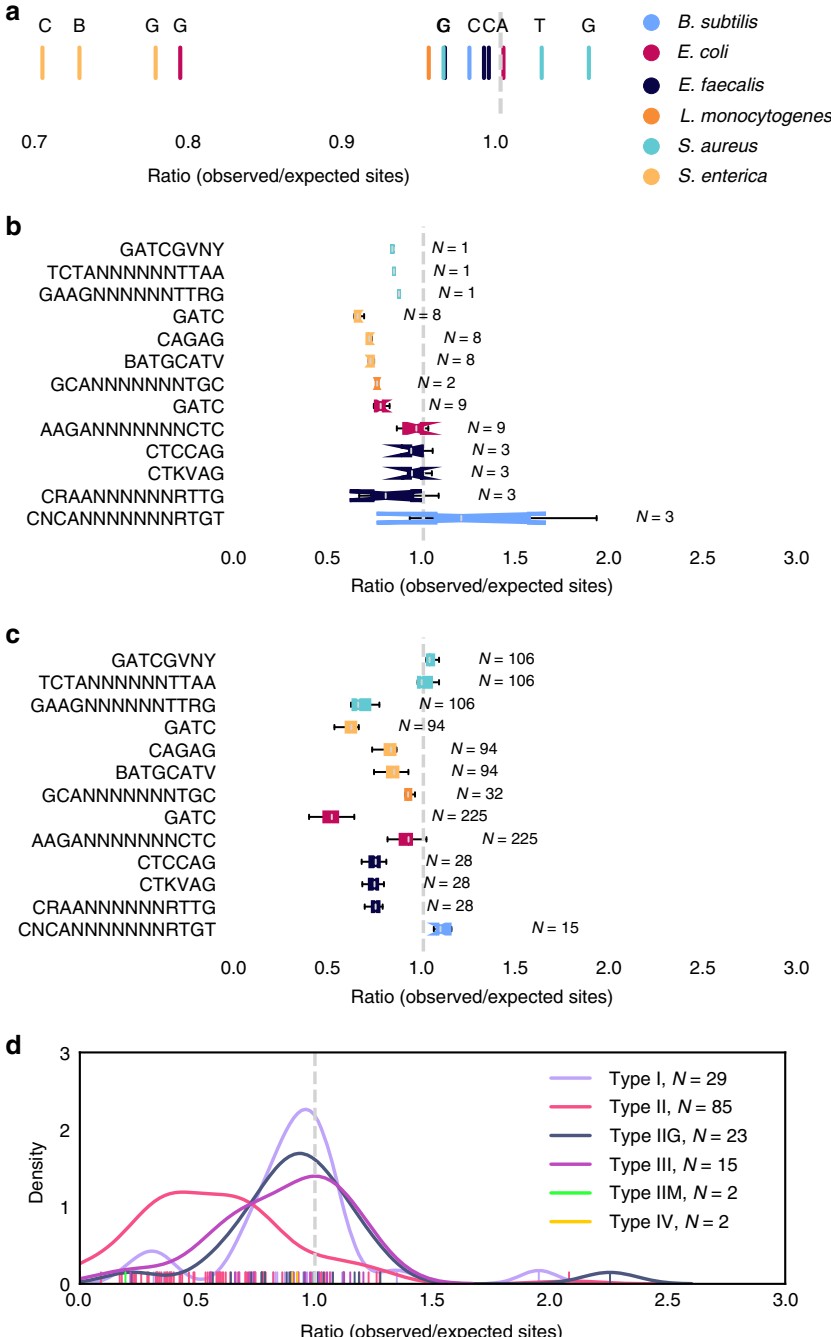

**Fig. 5** Methylated motif depletion. Ratio of observed over expected motif counts calculated by maximal order Markov model **a** in assembled genomes, colored by species and labeled by first nucleotide where there were multiple motifs for a species, **b** in predicted prophages within assembled genomes, and **c** in viruses associated with each species. **d** The distributions of ratios for different types of restriction-modification system across viruses associated with different species in the REBASE database. Boxplot center lines show medians, notches confidence intervals, and whiskers 1.5x interquartile range

the genome for testing (though we found no change in accuracy with separation solely by experiment).

Nanopore reads were aligned to the genome from PacBio using GraphMap[48], although tests with BWA-MEM[49] did not show any effect on mCaller performance. We used an existing program, nanopolish eventalign[21,50], to re-align the R9 current data to reference sequences. Average read quality was added as an additional classification feature to better control for variation among experiments. By default, we did not classify observations with skips. All binary classification models were tested using the python package scikit-learn[51]. Parameters for the classifiers were adjusted using random grid search and 5-fold cross-validation. For the random forest model, grid search parameters were {"bootstrap": [True,False], "n_estimators": [40,50,60,100], "max_depth": [5,8,10,12], "max_features": [1–4], "min_samples_leaf": [1–3,10]}, with final

parameters {bootstrap = True, n_estimators = 50, max_depth = 10, max_features = 4, min_samples_leaf = 2}. For the neural network, grid search parameters were {"hidden_layer_sizes": [(4,4,4,4,4),(100),(100,100,100), (100,100,100,100)], "alpha": [0.0001, 0.001, 0.01], "activation": ["tanh", "relu", "logistic", "identity"]} and final parameters {hidden_layer_sizes = (100), alpha = 0.001, learning_rate = "adaptive", activation = "tanh"}. The logistic regression grid search parameters were all valid combinations of "solver": ["liblinear", "newton-cg", "lbfgs", "sag"], "penalty": ["l1", "l2"]}. We did not see any differences and kept the default parameters {solver = "liblinear", multi_class = "ovr", penalty = "l1"}. The naïve Bayes classifier parameters were the defaults for the GaussianNB classifier: {priors = None, var_smoothing = 1e −09}. The final parameters for each algorithm were used to test on a second dataset.

Model accuracy was calculated as $\frac{\#\text{ sites called correctly}}{\#\text{ total sites}}$, with equal weight given to known methylated and unmethylated sites. Area under the curve was calculated using the roc_auc_score function from the python package scikit-learn by sweeping over estimated probabilities of methylation for single-read predictions, or the fraction of reads with estimated probability of methylation ≥50% for multi-read predictions.

**Reference materials**. Data was generated on a variety of sequencing platforms for strains in a reference community (summary in Supplementary Table 3). Eight strains from the ZymoBIOMICS Microbial Community Standard were individually sequenced on the PacBio RSII and Sequel sequencer at two sites (FDA-ARGOS and University of Florida). PacBio reads were assembled using HGAP v2 where available and putative adenine methyltransferase genes annotated using RAST via PATRIC[52]. We did not have PacBio data for two strains because the DNA provided failed initial quality control. For *L. fermentum*, an assembly was generated using nanopore and Illumina iSeq100 data (further described in the legend for Supplementary Table 4), and for *C. neoformans*, an assembly was generated using Canu for nanopore data alone[53]. Bacterial assemblies were completed as one or two contigs (Supplementary Table 4). To verify the accuracy of the assemblies, we used the software package pyani (github.com/widdowquinn/pyani), which calculates Pearson's correlations of TETRA ANI values between assembled contigs and genomes in the NCBI database. For prokaryotes, similarity of contigs to both chromosomes and plasmids was considered; for eukaryotes, the similarity of the whole genomes was used. Pearson's *r* was 0.99866 or higher for all identified pairs of genomes. Completeness of each assembly was assessed with BUSCO, or Benchmarking Universal Single-Copy Orthologs[54] (Supplementary Table 5). Since different species have varying genetic similarity to the BUSCO database, genome completeness was compared between the assemblies and the genomes of the identified related strains. All assemblies had high scores, with percentages of total identified BUSCOs from 96.2 to 100%. The scores for the prokaryotic assemblies were on par with those for previously published genomes while the eukaryotic assemblies had lower scores. Most genome lengths matched published strains well, except for *C. neoformans*, for which the assembly was longer than the reference. Reads sequenced from the Zymo pooled community aligned slightly, but not significantly, better to our assemblies than the closest related strains (Supplementary Figure 7).

Modifications and motifs called with both SMRT Tools v2.3.0 and an updated version, v.5.0.1. Results were similar between the two versions although the newer version detected fewer false positive motifs and predicted a motif for *L. monocytogenes* that combined those found by the older version. We present the newer results in the main text and summarize all motifs and non-motif sites for which the modified base was an adenine in Supplementary Table 2. Results differed when we specified m6A detection rather than including all adenines detected as modified bases; the former strategy can increase precision at a cost to sensitivity. Six strains were resequenced on a PacBio Sequel and motifs again identified using SMRT Tools v.5.0.1.

The MethylFlash m6A DNA Methylation ELISA Kit from Epigentek was used to compare overall percent methylation, with two replicates generated for all ten Zymo Community strains and the average results for each species compared to percent methylation based on PacBio motif sites (all motifs in Fig. 3 were included in the calculation except for ANARA**G**TANYR, which was not recaptured in the Sequel data).

Nanopore data was generated by multiplexing strains and sequencing using a 1D ligation kit and R9.5 flowcell. Reads were basecalled using albacore (v1.2.2 and v2.0.2) and those with mean QV scores ≥10 were realigned and analyzed as described above (we have found that read quality scores are inconsistent between flowcell and basecaller versions and this threshold is not comparable to the high-quality reads described for R9 data, but that lower quality reads led to issues with nanopolish). A new neural network for R9.5 data was trained using *E. coli* positions identified by PacBio as methylated and corresponding to known m6A motifs, with a separate model trained for A**G** sites. The model was then tested on the nine other strains and a second run for the same *E. coli* species. Methylated motifs were compared to all known methylated motifs in REBASE[27] by taking the 40-mer surrounding a base identified as methylated and using AME[28] to compare to a random selection of control sites with <50% of reads called as methylated, with results described for *B. subtilis*, *C. neoformans*, and *L. fermentum*.

Tombo was run using the following commands:
$tombo resquiggle <fast5 directory> <reference genome fasta>--processes 8
$tombo test_significance --fast5-basedirs <fast5 directory>/
--alternate-bases 6 mA --statistics-file-basename sample
$tombo write_wiggles --wiggle-types fraction --statistics-filename sample.6 mA.tombo.stats

Precision-recall curves were generated by comparing percent methylation at each PacBio-validated motif position to percent methylation of either randomly selected unmethylated adenines in the same genome or the same motifs in *P. aeruginosa* (as we found no adenine methylation in the strain), as described in the legend for Fig. 4.

**MeDIP-seq**. A MeDIP-seq protocol was adapted from Koziol et al.[55]. RNase treatment was used only for *C. neoformans* and *B. subtilis* because of limited DNA.

Briefly, starting with 5 μg per sample, DNA was sheared for 80–120 s using a Covaris E220 sonicator with intensity 4, 175 W, 200 cycles/burst, at 7+/−2 degrees Celsius to generate fragments of approximately 200–300 base pairs. Three microgram of DNA were taken for immunoprecipitation and the rest set aside as an input control. For immunoprecipitation, DNA was incubated overnight in 200 μl 30 μg/μl bovine serum albumin stock solution (BSA), 200 μl 5× DIP buffer (0.5 ml 1 M Tris-HCl, pH 7.4, 1.5 ml 5 M NaCl, 0.5 ml 10% vol/vol Igepal CA-630), and water to a final volume of 1 ml, with 2.5 μl of 1 μg/μl anti-m6A antibody stock (SYSY 202 003). Hundred microliter magnetic protein A beads (Invitrogen) per sample were prepared by washing three times in 1 ml 1× DIP buffer and incubating overnight in 105 μl 1× DIP buffer and 105 μl BSA. Two hundred microliter of the bead suspension were added to each sample and the mixture incubated for 90 min, then washed four times in 1× DIP buffer, resuspended in 200 μl DIP elution buffer (45 μl 5× DIP buffer, 75 μl 20 mM N6-methyladenosine 5′-monophosphate sodium salt stock, 105 μl water), and placed at 42 °C for 1 h at 1400 RPM. To precipitate, the supernatant was transferred to a new tube and 300 μl water, 2 μl GlycoBlue (Invitrogen), 50 μl 3 M NaOAc, and 500 μl isopropanol added, then frozen at −80 °C, centrifuged for 30 min, washed twice in 70% ethanol, and resuspended in 1× TE.

The NEBNext Ultra I DNA Library Prep Kit for Illumina (E7645) with NEBNext Multiplex Oligos for Illumina (E6440) was used to prepare immunoprecipitated samples and input controls for sequencing. Reads were aligned to their appropriate genomes using BWA-MEM[56], and peaks called using MACS2 (--nomodel)[57]. Motif or non-motif sites that overlapped with peaks were considered m6A methylated for comparison between MeDIP-seq and single-molecule methods.

**Bisulfite sequencing**. Libraries for all ten strains from the Zymo BIOMICs controls and the *E. coli* K12 strain were prepared using the TruSeq DNA Methylation kit from Illumina 100 bp paired end reads generated on an Illumina HiSeq 2500. Reads were mapped using Bismark[58].

**REBASE canonical catalytic sites**. A list of all m6A methyltransferases was downloaded from REBASE (http://rebase.neb.com/cgi-bin/mtypelist?m6+f) and their associated unique protein sequences searched for DPPY and NPPY motifs.

**Phage motif underrepresentation**. Prophage genomes within our assemblies were identified using PHASTER[59]. We used all annotated prophage sequences, from incomplete to intact, for motif analysis. Additional phages that target the same species were identified using the GenomeNet Viral-Host Database[42]. Other species and methyltransferases, labeled by type, were taken from Rusinov et al.[13] and then associated with viruses using the same database. Only dsDNA viruses were considered as potential targets for RM systems. The maximal order Markov expected count was calculated as $\frac{N(s_{1:n-1})N(s_{2:n})}{N(s_{2:n-1})}$ where $N$ is the count for a submotif $s_x$ of a motif with length $N$. The Kr ratio was calculated as described by Rusinov et al.[13]—see Supplementary Information for details. In Fig. 5, density curves were plotted for all restriction enzyme types associated with at least fifteen motifs.

**Code availability**. The python 2.7 package is open source and available at github.com/al-mcintyre/mcaller. Scripts used for analysis and figure generation are available at https://github.com/al-mcintyre/mCaller_analysis_scripts.

**Reporting summary**. Further information on experimental design is available in the Nature Research Reporting Summary linked to this article.

## Data availability

Sequence data (fastq, fast5, and PacBio files) have been submitted to NCBI SRA with bioproject number PRJNA477598. Regulatory-grade genomes for the Zymo microbial community are also available in the FDA-ARGOS repository at PRJNA231221 under sample names FDAARGOS_606 to FDAARGOS_612. All other data are available from the authors upon reasonable request.

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

## Acknowledgements

We acknowledge Sneha Somasekar, Aaron Burton, Sarah Castro-Wallace, Kristen John, Sarah Stahl, and Sheng Li for generating some of the initial nanopore datasets used in testing an earlier model. We also thank Ryan Kemp and Shuiquan Tang at Zymo for providing DNA, Rashad Ahmed and Douglas Duckworth at PBTech for assistance installing PacBio SMRT tools, and the Genomics and Epigenomics Cores at Weill Cornell for preparing bisulfite sequencing libraries and Illumina sequencing. We would like to thank the National Science and Engineering Research Council, the Starr Cancer Consortium (I9-A9-071), the Bert L and N Kuggie Vallee Foundation, the WorldQuant Foundation, The Pershing Square Sohn Cancer Research Alliance, NASA (NNX14AH50G, NNX17AB26G), the National Institutes of Health (R25EB020393, R01NS076465, R01AI125416, R01ES021006, 1R21AI129851, 1R01MH117406), the Bill

and Melinda Gates Foundation (OPP1151054), the Leukemia and Lymphoma Society (LLS) grants (LLS 9238-16, Mak, LLS-MCL-982, Chen-Kiang).

## Author contributions

A.B.R.M. and C.E.M. conceived the study. A.B.R.M. developed and tested the model and prepared nanopore and MeDIP-seq libraries. A.B.R.M. and N.A. analyzed PacBio data. C. Y.C. contributed nanopore data using in training and testing. H.S. contributed RSII PacBio data. K.G., N.A. and D.B. generated and evaluated assemblies. A.B.R.M. and C.E. M. wrote the manuscript. All authors read and edited the manuscript.

## Additional information

**Competing interests:** Zymo donated DNA for the purpose of the study. C.E.M. is a cofounder and board member for Biotia and Onegevity Health, as well as an advisor or compensated speaker for Abbvie, Acuamark Diagnostics, ArcBio, BioRad, DNA Genotek, Genialis, Genpro, Karius, Illumina, New England Biolabs, QIAGEN, Whole Biome, and Zymo Research. A.B.R.M. received a reimbursement for intracity travel to speak at an ONT community meeting. The remaining authors declare no competing interests.

