## [Peer Review File · Nature Communications]

Reviewers' Comments:

Reviewer #1:

Remarks to the Author:

The paper presents a new approach to detection of m6A methylation in data produced by Oxford Nanopore. In the first part of the paper, the authors describe a new tool called mCaller, which is based on modern machine learning techniques. Authors examined several possibilities, and they have chosen a single-layer neural network as their final approach.

In the second part, authors produced a new data set based on eight strains from the ZymoBIOMICS Microbial Community Standards, using a variety of technologies to characterize methylation. Based on their new approach and also using established tools, they performed an in-depth analysis of m6A methylation patterns on this standard. My expertise is mainly computational, so I am unable to judge novelty of this part. However, from the analysis, it seems clear that this data set is a very comprehensive resource for studying m6A methylation and if released in its entirety to the community, it would help greatly to further develop methods for methylation detection using a variety of technologies. Unfortunately, I was unable to verify whether the data were indeed released in their entirety, since the authors claim these to be a part of quite substantial bioproject at NCBI (FDA-ARGOS) and locating the files relevant to this study within this bioproject is somewhat difficult.

I believe this work is substantial, and of great interest. However, I do have certain concerns that I believe should be addressed in the final version of the manuscript.

>> p. 6: "Testing on a second data set (...) the model achieved 81% accuracy"

My main complaint with the paper is that testing of the model is not sound, as pointed to by the quoted paragraph on p. 6. Authors decided to use one data set produced on *E. coli* as a training set, and another data set produced using the same techniques on the same *E. coli* as a testing set. This is a mistake: In this scenario, the model that would instead of methylation learn to distinguish particular sequence contexts in the *E. coli* genome would also perform extremely well, and in fact there seems to be such an element at play here, since authors point out:

>> p.12: "While this method still has limitations, particularly for sequence contexts on which a model hasn't been trained (...)"

In other words, the model does not perform as well in species other than *E. coli*. One obvious way of fixing this issue would be to use e.g. half of the *E. coli* genome for training and the second half, previously unseen by the model, for testing.

>> p.12: "Our program is the first designed for training using orthogonally validated positions, rather than artificially methylated or unmethylated reads."

>> p.14: "methylated positions (were) identified through differences between measured and expected interpulse duration using SMRTPortal (...)"

The two quotes above point to another problem from the machine learning point of view. In both training and evaluation, the authors use methylation detection pipeline based on PacBio data and treat its result as the ground truth. However, as far as I can tell, the PacBio pipeline is far from perfect at identifying methylated sites (though it is difficult to find studies on the accuracy of this prediction method). Can this mean that, rather than learning detecting methylation sites, the model learns to imitate PacBio data, including its errors? Is this a simpler or a more difficult problem?

Also, there are multiple studies (Westphal et al. 2016 *mSystems*; Kahramonoglou et al. 2012 *Nat Commun*) pointing out that certain methylation patterns may be different in different phases. Were

there any steps taken to ensure that in all data sets, the DNA extraction was done under the same conditions?

>> p.11: "We show here that m6A (...) can be detected (...) as individual DNA molecules travel through a nanopore, with a new method that can improve detection compared to existing models."

In multiple places (quote on p. 11 being one of them, the title of the paper starting with "Single-molecule detection" being the second prominent example), paper implies that they can accurately detect methylation in individual DNA sequencing reads. The chief proof of this should be in Figure 4A, where very nice ROC curves are reported. However, reading the text carefully, it seems that these ROC curves are produced using 15x coverage consensus, rather than individual reads. In fact, ROC curves shown in Supplementary material (e.g. Supplementary Figure 2) seem to indicate much more modest results - I believe these were produced on single reads?

Tabular results are missing, AUC numbers are mostly presented only within the text and only ones that the authors chose to report. This makes judging the accuracy of the tool independently of the author's text very difficult.

>> p.2: "(...) and also demonstrate the ability to detect base modifications from data generated on the International Space Station"

>> p.5: "(...) and validated their improved functionality as a reference on a third platform (Illumina iSeq100)"

There is a tendency to overstate the results throughout the paper. The above quotes refer to interesting experiments, and I believe authors have done them, but I see no sections in Results devoted to analysis of this data.

>> p.6 and Figure 1: "We looked at current deviations (...) informative patterns were observed in several contexts (...)"

Based on Figure 1, I wouldn't dare to draw these conclusions about signal deviations (perhaps I would grant some conclusions on positions 3 and 5 in Figure 1A). In fact, Figure 1C seem to clearly indicate, that the shifts in a signal at most of the positions are statistically insignificant.

In conclusion, I do believe that this paper is significant and the results should be published. However, I also believe that substantial parts of the paper need to be rewritten and some computational experiments need to be redesigned to follow current best practices in machine learning. I also believe that the presentation of data should be clearer and the context of evaluation and comparison results should be much more apparent. The data should be available in such a way that they are easily located by other researchers wishing to reproduce the computational experiments. Finally, statements of results should be made carefully, not overstating the claims or referring to results that were apparently taken out of the paper.

Reviewer #2:

Remarks to the Author:

The manuscript "Single-molecule detection of N6-methyladenine in metagenome reference materials" by McIntyre et al reported a new algorithm mCaller for detecting modified base from Oxford Nanopore Technologies (ONT) nanopore sequencing data of metagenome. The authors compared and cross-validated m6A methylation sites detected by PacBio sequencing data and compared their method with another previously published algorithm.

Comment:

Detection of the DNA m6A modification in microbiome by single molecule sequencing data is of great importance for studying microbiome RM systems and other m6A functions. The major contribution of this manuscript is the application of neuron network classifier on current deviation of nanopore sequencing data to detect base modification. The authors also orthogonally benchmarked the methylation sites/motif data of reference materials.

1. However, the training data for the proposed algorithm mCaller is based on the m6A sites identified from PacBio dataset. Therefore, the m6A sites called by mCaller in this manuscript could potentially be influenced by unknown bias in PacBio dataset. In fact, recent studies suggest that PacBio approach severely overestimate m6A abundances. This could be a main problem. It will be interesting to apply alternative classifier Tombo to the same dataset analyzed by mCaller and compare the m6A sites called by Tombo with PacBio result in addition to mCaller-PacBio comparison as Tombo doesn't need training data.

2. In the comparison with alternative method Tombo (updated), mCaller was outperformed in one of three compared motifs. Therefore, it would be interesting to discuss the difference between the two models.

3. The authors also performed analysis of the m6A motifs in RM systems. But this analysis doesn't depend on the discovery made by the newly proposed algorithm.

Minor comment:

Color labeling of figure 4A is confusing. There are six curves but only 5 labels.

Figure 5 doesn't display correctly. Only two of four plots are displayed in the PDF file.

Reviewer #3:

Remarks to the Author:

McIntyre and colleagues presented their python package mCaller to analyze Nanopore data and predict the methylation of genomic DNA. Specifically they tested 6mA and 5mC in different microbe species. The DNA methylation is important modification to regulate gene expression and involve in many different biological process, the work presented here indicated a different but new way to detect 6mA by utilizing Nanopore sequencing platform. However, there are certain concerns when I read the manuscript.

Concerns:

1. Generally the manuscript is lack of details for any operation in this project. Especially as one methodology paper, details is very important to help reader understanding each step and apply the method by following the instruction.

2. Authors used 4 binary classifiers to predict the methylation status, and annoyed neural network classifier prefaced the highest accuracy. Then all the manuscript start to use the NN model to generate predication. As I mentioned above, there is no any detail about tuning and optimization of these 4 classifiers. No details and explanation about feature selection for each model. Such unclear description made me strongly suspect the performance of NN model. Especially in S-f-2, random forest show very similar performance, author need to show much more evidence to illustrate the performance of each model and the reason to choose NN model.

3. Authors also predict the 5mC by using published data, it is a good test and an interesting part. I

got confused by S-fig-3-A, is that a typo "Adenine Position in 6-mer" since author evaluated cytosine in this panel. Author annoyed the accuracy at 77%, how this accuracy calculated? Authors have to detailed describe what they did.

4. In page 9, authors stated 'sites marked as methylated by PacBio that did not correspond to motifs were called as methylated only at low rates by mCaller'. Is this mean the motif or nucleotide grammar is more important the methylated signal (currents) in mCaller? Is there any validation?

5. In page 9, authors mentioned "a consistent decrease with adenine at 5th position was found for GATCGVNY in *S. aureus*, indicated PacBio may underestimated methylation at this motif". Using Nanopore data to evaluate the PacBio signal is a great idea, it will help us to see the platform-related bias. However, such announcement also need validation, since mCaller used PacBio data in training step. Is this a problem of training or the reality?

6. In page 5, author announced they validated the improved functionality as a reference on iSeq100. Where is the validation? It is easy to understand a better genome assembly can be validated by iSeq100, but how to validate m6A epigenomes?

Reviewers' comments:

Reviewer #1 (Remarks to the Author):

The paper presents a new approach to detection of m6A methylation in data produced by Oxford Nanopore. In the first part of the paper, the authors describe a new tool called mCaller, which is based on modern machine learning techniques. Authors examined several possibilities, and they have chosen a single-layer neural network as their final approach.

In the second part, authors produced a new data set based on eight strains from the ZymoBIOMICS Microbial Community Standards, using a variety of technologies to characterize methylation. Based on their new approach and also using established tools, they performed an in-depth analysis of m6A methylation patterns on this standard. My expertise is mainly computational, so I am unable to judge novelty of this part. However, from the analysis, it seems clear that this data set is a very comprehensive resource for studying m6A methylation and if released in its entirety to the community, it would help greatly to further develop methods for methylation detection using a variety of technologies. Unfortunately, I was unable to verify whether the data were indeed released in their entirety, since the authors claim these to be a part of quite substantial bioproject at NCBI (FDA-ARGOS) and locating the files relevant to this study within this bioproject is somewhat difficult.

Thank you for your comments. We submitted all of our nanopore, PacBio, and illumina data for the Zymo microbial community strains to NCBI ([PRJNA477598](https://www.ncbi.nlm.nih.gov/bioproject/PRJNA477598)) and it will be available on publication; also FDA-ARGOS is hosting additional assemblies for these strains on their site here:

https://www.ncbi.nlm.nih.gov/biosample/?term=FDAARGOS_606

https://www.ncbi.nlm.nih.gov/biosample/?term=FDAARGOS_607

https://www.ncbi.nlm.nih.gov/biosample/?term=FDAARGOS_608

https://www.ncbi.nlm.nih.gov/biosample/?term=FDAARGOS_609

https://www.ncbi.nlm.nih.gov/biosample/?term=FDAARGOS_610

https://www.ncbi.nlm.nih.gov/biosample/?term=FDAARGOS_611

https://www.ncbi.nlm.nih.gov/biosample/?term=FDAARGOS_612

https://www.ncbi.nlm.nih.gov/biosample/?term=FDAARGOS_613

The *E. coli* K12 data was previously published as part of the Castro-Wallace, et al. (2018) paper and we have added the FAST5 files to [PRJNA477598](https://www.ncbi.nlm.nih.gov/bioproject/PRJNA477598).

I believe this work is substantial, and of great interest.

We thank the reviewer for the positive comments.

However, I do have certain concerns that I believe should be addressed in the final version of the manuscript.

>> p. 6: "Testing on a second data set (...) the model achieved 81% accuracy"

My main complaint with the paper is that testing of the model is not sound, as pointed to by the quoted paragraph on p. 6. Authors decided to use one data set produced on *E. coli* as a training set, and another data set produced using the same techniques on the same *E. coli* as a testing set. This is a mistake: In this scenario, the model that would instead of methylation learn to distinguish particular sequence contexts in the *E. coli* genome would also perform extremely well, and in fact there seems to be such an element at play here, since authors point out:

>> p.12: "While this method still has limitations, particularly for sequence contexts on which a model hasn't been trained (...)"

In other words, the model does not perform as well in species other than *E. coli*. One obvious way of fixing this issue would be to use e.g. half of the *E. coli* genome for training and the second half, previously unseen by the model, for testing.

We agree that this is a limitation of the testing on the *E. coli* data and included the Zymo microbial standards to determine the extent to which our models would be biased towards specific contexts given limited training data. We did split the *E. coli* genome by 11-mer for 5-fold cross-validation, but since the motifs are still mainly GATC across the whole genome, training and testing on alternate 11-mers (or splitting the genome in two) is not sufficient to remove the sequence motif bias learned by the model and there was no difference in accuracy. We have rerun the *E. coli* training and testing using separate halves of the genome and present the revised results. However, we have emphasized in the discussion that testing with a single species that expresses one or two methyltransferases, regardless of splits in the genome, is a start, but not an adequate evaluation on its own of any model.

>> p.12: "Our program is the first designed for training using orthogonally validated positions, rather than artificially methylated or unmethylated reads."

>> p.14: "methylated positions (were) identified through differences between measured and expected interpulse duration using SMRTPortal (...)"

The two quotes above point to another problem from the machine learning point of view. In both training and evaluation, the authors use methylation detection pipeline based on PacBio data and treat its result as the ground truth. However, as far as I can tell, the PacBio pipeline is far from perfect at identifying methylated sites (though it is difficult to find studies on the accuracy of this prediction method). Can this mean that, rather than learning detecting methylation sites, the model learns to imitate PacBio data, including its errors? Is this a simpler or a more difficult problem?

We too were unable to find studies that evaluate the accuracy of PacBio, mainly because there are no other high-throughput gold standard sequencing methods to locate m⁶A at single base resolution. Our MeDIP-Seq data and WGBS data for this paper have help our work and will also be useful for future studies (below).

The results from PacBio used for training align with what we know from literature on m⁶A biology: (1) it is well established that the vast majority of GATC motifs in *E. coli* expressing Dam are methylated (see Marinus and Casades review, 2009, Westphal et al., 2016), and (2) the M.EcoKI target motifs

AACNNNNNNGTGC/GCACNNNNNNGTT are also highly methylated in *E. coli* K12 as otherwise genomic DNA would be cleaved by the associated restriction enzyme (see for example Murray review, 2000, O'Neill, et al., 2001, Su et al., 2005). While updating our training and testing sets to divide across the genome, we also introduced the explicit condition that m⁶A sites be associated with these known motifs.

We were less confident that PacBio correctly called the motifs from other species. We have therefore added an immunoprecipitation-based assay (MeDIP-seq) as further validation, presented in Figure 3 and Supplementary Table 2. In general, the motif sites identified by PacBio are confirmed by MeDIP-seq, with the exception of GATCGVNY in *S. aureus*. mCaller also predicts m⁶A at GATCGVNY sites, but this is likely a function of the fact that both ONT and PacBio rely on changes in signal that may not discriminate as well as immunoprecipitation between different modified bases, rather than an imitation of a PacBio error (see Supplementary Figure 4, showing the current differences). We have expanded our discussion of this important point.

Also, there are multiple studies (Westphal et al. 2016 mSystems; Kahramonoglou et al. 2012 Nat Commun) pointing out that certain methylation patterns may be different in different phases. Were there any steps taken to ensure that in all data sets, the DNA extraction was done under the same conditions?

The same aliquots of DNA were used for PacBio and nanopore sequencing. DNA is also routinely harvested for the microbial community DNA at the same time points in growth for different species. From personal correspondence with Zymo: "The cultivation of each strain starts from one single colony on an agar plate. One single colony was used to inoculate 20 ml culture and cultivate for 12 hours. This culture then served as an inoculum for subsequent scale-up cultivation with 0.5% (v/v) inoculum size. The 2nd cultivation phase varied from 24 to 48 hours depending on the growth rate of each culture. For fast growing *E. coli* and *Salmonella*, the cultivation time is 24 hours. For slow growing *Lactobacillus fermentum*, the cultivation time is 30 hours. For *Cryptococcus neoformans*, the cultivation time is 48 hours." Cultivation thus occurs at the exponential or stationary phase, and is specifically and consistently timed depending on species. We have adapted some of this text for inclusion in the methods.

>> p.11: "We show here that m⁶A (...) can be detected (...) as individual DNA molecules travel through a nanopore, with a new method that can improve detection compared to existing models."

In multiple places (quote on p. 11 being one of them, the title of the paper starting with "Single-molecule detection" being the second prominent example), paper implies that they can accurately detect methylation in individual DNA sequencing reads. The chief proof of this should be in Figure 4A, where very nice ROC curves are reported. However,

reading the text carefully, it seems that these ROC curves are produced using 15x coverage consensus, rather than individual reads. In fact, ROC curves shown in Supplementary material (e.g. Supplementary Figure 2) seem to indicate much more modest results - I believe these were produced on single reads?

The ROC curves in both Figure 2 and Supplementary Figure 2 show per-read accuracy (Supplementary Figure 2 shows the basis for our selection of the model used in Figure 2 and for the rest of the paper through comparison to less accurate variations). We do use multiple reads for the comparison to Tombo in Figure 4, as Tombo estimates methylation per-position. We have clarified this in the text and figure legend.

We intended to indicate in the title that the sequencing platforms used and compared (nanopore and PacBio single-molecule real-time sequencing) are both based on single-molecule readings and have therefore changed the title to “Single-molecule sequencing detection of ...”, which we agree is more inclusive of some of our analyses.

Tabular results are missing, AUC numbers are mostly presented only within the text and only ones that the authors chose to report. This makes judging the accuracy of the tool independently of the author's text very difficult.

Very sorry for the formatting issue.

>> p.2: "(...) and also demonstrate the ability to detect base modifications from data generated on the International Space Station"

>> p.5: "(...) and validated their improved functionality as a reference on a third platform (Illumina iSeq100)"

There is a tendency to overstate the results throughout the paper. The above quotes refer to interesting experiments, and I believe authors have done them, but I see no sections in Results devoted to analysis of this data.

We agree and have removed the reference to previously published data from Castro-Wallace, et al., 2018, which was used only in early development for this method as it was generated from an obsolete version of the ONT flowcell and chemistry (R7.3). The iSeq data did not validate any m⁶A detection (for our new MeDIP-seq data, a single species was tested on the iSeq) and was used for only three of the assemblies. We have clarified this throughout.

>> p.6 and Figure 1: "We looked at current deviations (...) informative patterns were observed in several contexts (...)"

Based on Figure 1, I wouldn't dare to draw these conclusions about signal deviations (perhaps I would grant some conclusions on positions 3 and 5 in Figure 1A). In fact, Figure 1C seem to clearly indicate, that the shifts in a signal at most of the positions are statistically insignificant.

We hypothesized predictions would be possible based on information in the signal deviations observed, and have duly rephrased the sentence.

In conclusion, I do believe that this paper is significant and the results should be published. However, I also believe that substantial parts of the paper need to be rewritten and some computational experiments need to be redesigned to follow current best practices in machine learning. I also believe that the presentation of data should be clearer and the context of evaluation and comparison results should be much more apparent. The data should be available in such a way that they are easily located by other researchers wishing to reproduce the computational experiments. Finally, statements of results should be made carefully, not overstating the claims or referring to results that were apparently taken out of the paper.

Thank you for your comments.

Reviewer #2 (Remarks to the Author):

The manuscript “Single-molecule detection of N6-methyladenine in metagenome reference materials” by McIntyre et al reported a new algorithm mCaller for detecting modified base from Oxford Nanopore Technologies (ONT) nanopore sequencing data of metagenome. The authors compared and cross-validated m6A methylation sites detected by PacBio sequencing data and compared their method with another previously published algorithm.

Comment:

Detection of the DNA m6A modification in microbiome by single molecule sequencing data is of great importance for studying microbiome RM systems and other m6A functions. The major contribution of this manuscript is the application of neuron network classifier on current deviation of nanopore sequencing data to detect base modification. The authors also orthogonally benchmarked the methylation sites/motif data of reference materials.

1. However, the training data for the proposed algorithm mCaller is based on the m6A sites identified from PacBio dataset. Therefore, the m6A sites called by mCaller in this manuscript could potentially be influenced by unknown bias in PacBio dataset. In fact, recent studies suggest that PacBio approach severely overestimate m6A abundances. This could be a main problem. It will be interesting to apply alternative classifier Tombo to the same dataset analyzed by mCaller and compare the m6A sites called by Tombo with PacBio result in addition to mCaller-PacBio comparison as Tombo doesn't need training data.

As noted in our response to Reviewer 1 above, we re-trained based on established highly methylated motifs (GATC - Marinus and Casadesus review, 2009, AACNNNNNGTGC/GCACNNNNNGTT - Westphal et al., 2016, Murray review, 2000, O'Neill, et al., 2001, Su et al., 2005) to avoid bias from using PacBio as the source of training sites, although keep PacBio as an extra filter for confirmation.

Our comparison to Tombo in Figure 4 is based on motifs predicted by PacBio. Since Tombo has not yet been formally published, we will leave it up to its authors whether they would like to use our PacBio data for further evaluations.

2. In the comparison with alternative method Tombo (updated), mCaller was outperformed in one of three compared motifs. Therefore, it would be interesting to discuss the difference between the two models.

We have added to the discussion of differences between the models, but are unfortunately limited by a lack of transparency, as the main source of information available is the 2016 bioRxiv publication.

3. The authors also performed analysis of the m⁶A motifs in RM systems. But this analysis doesn't depend on the discovery made by the newly proposed algorithm.

True, although the better validated m⁶A motifs are, the more meaningful these analyses become. Also, we believe these RM system metrics can help guide new metrics for genome and epigenome references materials, like those used in this paper.

Minor comment:

Color labeling of figure 4A is confusing. There are six curves but only 5 labels.

Figure 5 doesn't display correctly. Only two of four plots are displayed in the PDF file.

Thank you for your comments; we have fixed the hidden label in Figure 4 and the pdf for Figure 5.

Reviewer #3 (Remarks to the Author):

McIntyre and colleagues presented their python package mCaller to analyze Nanopore data and predict the methylation of genomic DNA. Specifically they tested 6mA and 5mC in different microbe species. The DNA methylation is important modification to regulate gene expression and involve in many different biological process, the work presented here indicated a different but new way to detect 6mA by utilizing Nanopore sequencing platform. However, there are certain concerns when I read the manuscript.

Concerns:

1. Generally the manuscript is lack of details for any operation in this project. Especially as one methodology paper, details is very important to help reader understanding each

step and apply the method by following the instruction.

The updated methods are publicly available on github along with a test example and up-to-date instructions on how to run the tool. We have also addressed the comments below to help clarify the methods, but please let us know if you identify any other specific details that are missing.

2. Authors used 4 binary classifiers to predict the methylation status, and a neural network classifier prefaced the highest accuracy. Then all the manuscript start to use the NN model to generate predication. As I mentioned above, there is no any detail about tuning and optimization of these 4 classifiers. No details and explanation about feature selection for each model. Such unclear description made me strongly suspect the performance of NN model. Especially in S-f-2, random forest show very similar performance, author need to show much more evidence to illustrate the performance of each model and the reason to choose NN model.

The features for all models were the same, and are described in Figure 2: the current deviations from model values associated with sequences surrounding methylated bases and read quality scores. The only variations to features were the number of current deviations considered, as shown in Supplementary Figure 2. We chose to look at current deviations based on their potential to differentiate adenine and methylated adenine, illustrated in Figure 1, and looked to select the best possible model (NN, RF, LR, or NBC) to predict methylation based on these current deviations. We note in the methods that the models were each tuned using a grid search and have added additional details on the parameters explored.

We agree that the random forest model performed comparably and leave it as an option for users looking to train their own models, but also note that it ran more slowly than the neural network (almost 5X the time it took for the neural network to classify our E. coli K12 test set using 8 threads, although future optimization may be possible). We have also added the accuracy for the random forest and neural network models from Supplementary Table 1 to the text for ease of comparison.

3. Authors also predict the 5mC by using published data, it is a good test and an interesting part. I got confused by S-fig-3-A, is that a typo “Adenine Position in 6-mer” since author evaluated cytosine in this panel. Author annoyed the accuracy at 77%, how this accuracy calculated? Authors have to detailed describe what they did.

We have fixed the typo; thank you for catching this. The accuracy is evaluated in the same way as for the m⁶A, # sites correct/#total sites, with equal weighting given to methylated and unmethylated test sites. We have added this explanation to the Methods section.

4. In page 9, authors stated ‘sites marked as methylated by PacBio that did not correspond to motifs were called as methylated only at low rates by mCaller’. Is this

mean the motif or nucleotide grammar is more important the methylated signal (currents) in mCaller? Is there any validation?

It suggests that the sites are not truly methylated at most of these sites, given mCaller is able to detect m⁶A across a range of untrained sequence contexts. Motifs are not taken into account at all in mCaller predictions, although we note that current signal deviations differ according to the surrounding sequence. Based on previous studies, m⁶A is thought to occur at specific motifs except in rare cases (see reference 31, Murray et al., 2017). We have now added MeDIP-seq data to Figure 3C and the text on page 9 to better make this point.

5. In page 9, authors mentioned “a consistent decrease with adenine at 5th position was found for GATCGVNY in *S. aureus*, indicated PacBio may underestimated methylation at this motif”. Using Nanopore data to evaluate the PacBio signal is a great idea, it will help us to see the platform-related bias. However, such announcement also need validation, since mCaller used PacBio data in training step. Is this a problem of training or the reality?

There are limited methods to detect m⁶A and only two based on single-molecule sequencing. MeDIP-seq data is now included in Figure 3A and Supplementary Table 2 as additional validation for our training and test motif sites.

GATCGVNY turned out to be an interesting case. Current differences in ONT (Supplementary Figure 4) and IDP differences in PacBio strongly suggested that a modification was present but MeDIP-seq (for m⁶A) did not pick up m⁶A. Bisulfite sequencing (for m⁴C, m⁵C, or hm⁵C) was able to confirm it was a cytosine modification. The ambiguity in single-molecule sequencing results demonstrates a weakness of both PacBio and nanopore methylation detection in discriminating possible sources of small signal disruptions.

6. In page 5, author announced they validated the improved functionality as a reference on iSeq100. Where is the validation? It is easy to understand a better genome assembly can be validated by iSeq100, but how to validate m⁶A epigenomes?

We have removed this reference to iSeq100, which was used to improve genome assemblies for three of the ten strains and to evaluate the assemblies but not the methylomes. We apologize for the confusion and thank you for your comments.

Reviewers' Comments:

Reviewer #1:

Remarks to the Author:

The authors satisfactorily addressed my comments from the previous round.

Reviewer #2:

Remarks to the Author:

The authors have addressed most concerns I have.

Reviewer #3:

None

REVIEWERS' COMMENTS:

Reviewer #1 (Remarks to the Author):

The authors satisfactorily addressed my comments from the previous round.

Reviewer #2 (Remarks to the Author):

The authors have addressed most concerns I have.

We thank the reviewers for their comments. No further reviewer concerns to address.